# Risk factors associated with gall bladder cancer in high incidence areas in India: a systematic review protocol

Eliza K Dutta ![ORCID], Melissa Glenda Lewis, Sandra Albert

## ABSTRACT

**Introduction** Gall bladder cancer (GBC) is a lethal form of malignancy of the gastrointestinal tract with a unique geographical distribution. Cases are aggregated in the river basins of Ganga-Brahmaputra, in the north, east and north-east India, collectively termed as the 'high-risk' zone for GBC. Although some studies report high prevalence of typhoid infection linking with high burden of GBC in these regions, there is no systematic review of the factors associated with GBC in the high incidence areas. To address this gap, we are conducting a systematic review to identify and organise the factors associated with GBC in the high-risk zone of India.

**Methods and analysis** A systematic review of all observational studies that report a quantitative relationship between at least one risk factor for GBC among adults (>18 years) in the high-risk zone in India will be conducted. The databases PubMed-MEDLINE, CINAHL, EMBASE, Web of Science, Scopus, OpenGrey and Google Scholar published in English and after 1990 will be searched. This review will follow the Preferred Reporting Items for Systematic Reviews and Meta-Analyses recommendations. The primary outcome is GBC. If data permit, meta-analysis will be performed. Two independent reviewers will independently screen the articles, extract the data and assess the methodological quality of the studies.

**Ethics and dissemination** As this will be a systematic review without human participants' involvement, there will be no requirement for ethics approval. Findings will be disseminated widely through peer-reviewed publication and media, for example, conferences and symposia.

**PROSPERO registration number** CRD42021256673.

## Strengths and limitations of this study

► The proposed systematic review addresses a gap by providing a comprehensive search on risk factors for gall bladder cancer from the high-risk zone of India, which reports the third highest incidence in the world. Although sporadic narrative reviews exist, a systematic review on this topic is first from the region.

► The review strictly follows the Preferred Reporting Items for Systematic Reviews and Meta-Analyses guidelines.

► This review is limited to observational studies published in English.

► There is a potential for low and inconsistent quality in the studies as we anticipate many small studies that are not adequately powered.

## INTRODUCTION

Gall bladder cancer (GBC) is one of the most lethal forms of malignancy of the gastrointestinal tract with an overall survival <1 year.[1–4] Salient features of this cancer include: (a) non-specific presentation and asymptomatic progression,[5] and thus detection at a very late stage with poor prognosis; and (b) unique geographical distribution[3 5] with cases aggregated in river basin zones such as the Ganga-Meghna-Brahmaputra (GMB) plain.[6] Population-based cancer registry (PBCR) data from India report the highest incidence in Kamrup district (Assam), 16.2 and 7.9 per 100 000 women and men,[7] respectively, only next to rates reported from Chile and Korea.[8] The risk of incident GBC is seven times higher in the north and north-eastern states as compared with the south (Age-standardized rate of incidene per 100 000 women (ASR): 8.6–17.1 in north/north-eastern PBCRs vs 0.7–4.1 in southern PBCRs).[3 7 9] Of 29 Indian PBCRs reporting GBC, the top five incident areas are situated in the north-east India (Kamrup, Cachar and Dibrugarh districts in Assam, Papampure district in Arunachal Pradesh, Imphal west district in Manipur).[7] Evidence also indicates this risk remains the same even after migration from 'high-risk' north and north-eastern zones to the 'low-risk' southern regions (OR: 1.36, 95% CI: 1.02 to 1.82).[5 9] This sustained risk, therefore, is not explained by the already studied risk factors such as gallstones, typhoid infection, etc, but could be due to sustained effects of exposure to environmental factors[4] such as pesticides[10] and heavy metals,[11 12] especially at an early time point in life. Another possible explanation for sustained GBC risk in these populations could be genetic susceptibility. Familial relative risk of 3.15 with 23% heritability has

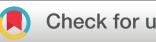

Indian Institutes of Public Health, Shillong, Meghalaya, India

**Correspondence to**
Dr Eliza K Dutta;
eliza.dutta@iiphs.org

been reported for GBC, indicating a significant contribution of genetic factors.[3]

Older age (>50 years), female gender, obesity (body mass index >25 kg/m$^2$), chronic *Salmonella typhi* and *Helicobacter bilis* infection, and gallstone, especially with stone size >3 cm and multiple stones are some of the established risk factors in the existing literature.[2 4 13 14] There is also evidence indicating a moderate to weak association between dietary patterns such as consumption of fried food, the interval between meals >8 hours, and exposure to carcinogen, coal/wood dust, and tobacco.[6 13 15] Reproductive factors such as multiparity, use of oral contraceptives, early menarche and late menopause have also shown association with GBC; however, the strength of this association remains unproven.[14 16] Although there is some evidence linking the prevalence of typhoid infection or exposure to arsenic, and geographical distribution of GBC in some parts of the world such as the Gangetic belt,[6 13] majority of these already established risk factors do not explain the unique geographical distribution of the disease, neither provide an understanding of the pattern of risk factors in the high incidence regions, for example, the north-eastern parts of the country, that report the highest incidence of GBC.[7]

Lack of concrete evidence is attributable to the fact that most of the reviews conducted to date have been (a) narrative reviews,[4 17] (b) not specific to population rather comprising studies across the world,[4 18] and (c) studies reporting associations based on post-surgical biospecimen included along with epidemiological studies.[19] There is a paucity of systematic evidence of risk factors that will enable us to understand the unique distribution of GBC. There is also a scarcity of literature on the pattern of risk factors specific to high burden areas in the world. Existing systematic reviews[20] report association with single risk factors (eg, size of gallstone, obesity) rather than providing evidence for population-specific risk factors.

As already stated, India, especially north and northeast, is one of the high incidence hotspots for GBC.[7] There is a paucity of systematically collated epidemiological evidence that would help understand the pattern of risk factors for GBC, and in turn explain for high burden in these regions. The north-eastern region of India is unique, home to different indigenous groups with unique sociocultural and traditional behavioural practices. For example, tobacco use is an integral cultural practice in the region,[21–23] tracing back to centuries, and the association between tobacco and the majority of cancers is a well-established phenomenon.[24] This is supported by high incidence of cancers of the mouth/oral cavity and nasopharynx in the region.[25] It is, therefore, necessary to study the patterns of risk factors in specific regions and populations to understand the distribution of the disease GBC in this case. As of now, we do not have enough evidence on risk factors explaining the high burden of GBC in India or north-east India. We, therefore, propose conducting a systematic review and meta-analysis of published literature to identify GBC risk factors of the study population (GMB belt in north, east and north-east India), which would help identify and organise the factors associated with GBC in the high-risk zone of India, helping develop effective prevention strategies.

## Objective
To identify and list the risk factors that are associated with GBC in high incidence areas in India.

## METHODS
### Standards
For the present protocol, we have followed the Preferred Reporting Items for Systematic Reviews and Meta-Analyses (PRISMA) guidelines.[26] This review is based on the methodology of the Cochrane handbook.[27] We will adhere to the PRISMA statement for the publication of the final review. We have registered the protocol to the International Prospective Register of Systematic Reviews database (No. CRD42021256673).[28]

### Eligibility criteria
All observational studies that include cohort, case–control and cross-sectional studies that report a quantitative relationship between at least one risk factor with GBC in the high incidence areas of India will be included. Articles that present only the incidence or prevalence of GBC will be excluded from this review. We will exclude studies that determined the effectiveness/efficacy of a treatment for GBC (randomised and non-randomised) unless they have assessed the risk factors. Reports from organisations, review papers, editorials, conference abstracts, research theses, qualitative studies, interviews, case series, case reports or studies that do not provide a quantitative relationship between the risk factors with GBC will be excluded. This review will be limited to articles in the English language published after 1990. We will also exclude studies reporting genetic associations with GBC.

### Participants
Studies published on adults (>18 years) with confirmed GBC in the high incidence areas in India will be included. In the present study, we define high incidence areas to be north, east and north-east India, comprising the Ganga-Brahmaputra belt.

### Disease
Adenocarcinoma of the gall bladder (International Classification of Disease, ICD 10-C23) only will be included in this review.

### Exposure
Risk factors associated with GBC.

### Comparator
Participants not having GBC will be the comparator group.

## Outcomes

The present study aims to identify the risk factors associated with GBC. Hence, the outcome is GBC.

## Patient and public involvement

There will be no patient or public involvement in this study, as it is based on secondary data.

## Information sources

A comprehensive literature search to identify all published and unpublished studies with no language restriction will be carried out. The electronic databases included in the search are PubMed-MEDLINE, EMBASE, Web of Science, Scopus, CINAHLplus, OpenGrey and Google Scholar. For each electronic database, a search strategy specific to that database will be developed. The search strategy for all the above databases is available in the online supplemental file 1.

## Searching other resources

We will follow a three-stage selection process for the final inclusion of studies in the review. In the first stage, two reviewers will assess each title for its appropriateness for inclusion in the review. If found inappropriate, then the title will be rejected, and all other titles will be moved to the second stage of selection. In the second stage, abstracts of titles will be obtained, and two reviewers will independently scrutinise all such abstracts. Here again, whenever both reviewers agreed to reject, such studies will be rejected, and the remaining studies will be obtained and reviewed by two authors independently. If both reviewers agreed to accept then those studies will be included, articles will be rejected when both reviewers agreed to reject. In case of disagreement between reviewers, a third reviewer will arbitrate the selection process. A PRISMA chart will be used to present the flow of the selection process.

## Data extraction

Data will be extracted using a predesigned and pretested proforma. The proforma will include the following: year of publication, authors, region, state, study design, study setting, information on the type participants, age group, gender, sampling technique, sample size, examined risk factor along with techniques used to measure each risk factor and its relationship with GBC (eg, incidence ratio/prevalence ratio/mortality ratio/OR/relative risk/HR). Information on all the confounding variables used for analysis, unadjusted and adjusted estimates will be extracted. Studies will be divided into two groups— adjusted or not adjusted for confounding factors. Data will be extracted from full-text articles by one reviewer and will be reviewed by a second reviewer. Disagreements, if any, will be discussed and will be drawn.

## Risk of bias

The risk of bias in non-randomised studies of interventions tool[29] will be used for assessing the risk of bias for the final set of articles on case–control, cohort and cross-sectional studies. We will treat studies that do not adjust risk factors for potentially confounding variables as 'high' risk of bias. Two reviewers will independently assess the risk of bias and conflicts will be resolved through consultation with a third reviewer. Cochrane risk of bias tool[30] will be used if interventional studies are obtained, although not likely.

## Data synthesis

First, we will provide a list of all confirmed risk factors mentioned in the included studies in a narrative format. A detailed summary of all the included studies will include information on authors, type of study design, participants, age, gender, region, sample size, the risk factor with a measure of association and primary findings including limitations. Second, an evaluation will be conducted if it is appropriate to perform a meta-analysis to assess the relationship of risk factors with GBC. Meta-analysis with random-effects model will be performed if there is a similarity in terms of the participants, study design and risk factors. The results will be expressed in relative risk, prevalence ratio and OR with 95% CIs when appropriate. Forest plots, $I^2$ statistic, $X^2$ test and $tau^2$ will be used to measure and assess heterogeneity among the included studies in each analysis. Meta-regression will be used to investigate heterogeneity if needed. An attempt will be made to contact study authors if data are inadequate or missing, and a record will be maintained on the amount of missing data with reasons. An assessment for publication bias will be made by creating a funnel plot only if there are at least 10 studies in the meta-analysis.

## Analysis of subgroups or subsets

If enough data are available, subgroup analysis will be performed by demographic factors, anthropometric factors, nutritional factors, reproductive factors, exposure to environmental factors, typhoid/enteric fever, dietary/cooking practices and geographical factors.

## Current study status

Review ongoing.

## Start date of the review

June 2021.

## Anticipated end date of review

September 2022.

## Ethics and dissemination

To our information, this will be the first study to synthesise results on the relationship between risk factors and GBC risk. The findings from the review will be helpful for future studies/reviews to compare the pattern of risk factors from other geographies/populations. These findings will be summarised and presented at conferences and through publication in a peer-reviewed journal. This study involves analysis of data from published literature and does not involve individual-level identifiable data.

Given this, there were no privacy concerns that required ethical approval.

**Acknowledgements** We acknowledge the India Alliance DBT-Wellcome Trust for funding this work under the 'Early Career Fellowship' in clinical research and public health. We also acknowledge the Indian Institute of Public Health, Shillong for providing necessary infrastructural support.

**Contributors** EKD conceptualised, designed, drafted the initial and reviewed the protocol. MGL planned data extraction and statistical analysis. SA defined the clinical concepts and reviewed the final protocol.

**Funding** India Alliance DBT-Wellcome Trust 'Early Career Fellowship' in clinical research and public health (grant no. IA/CPHE/19/1/504604).

**Competing interests** None declared.

**Patient consent for publication** Not required.

**Provenance and peer review** Not commissioned; externally peer reviewed.

**ORCID iD**
Eliza K Dutta http://orcid.org/0000-0002-2829-7059

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
