## [Reviewer comments · BMJ Open]

ARTICLE DETAILS

TITLE (PROVISIONAL)	Risk factors associated with gall bladder cancer in high incident areas in India: a systematic review protocol
AUTHORS	Dutta, Eliza; Lewis, Melissa; Albert, Sandra

VERSION 1 – REVIEW

REVIEWER	Patkar, Shraddha Tata Memorial Centre, Department of Surgical Oncology
REVIEW RETURNED	07-Nov-2021

GENERAL COMMENTS	none
------

REVIEWER	Nakagawa, Kei Tohoku University, Surgery
REVIEW RETURNED	07-Nov-2021

GENERAL COMMENTS	The protocol entitled, "Risk factors associated with gall bladder cancer in high incident areas in India: a systematic review protocol" presents an attempt to analyze risk factors for gallbladder cancer in a systematic review. I believe that it will be meaningful in research related to the epidemiology of carcinogenesis in specific regions and diseases. It is planned according to PRISMA-P to search for risk factors. I confirmed that it was properly planned with the checklist. On the other hand, I don't think there is much mention of sample size, the number of target factors, and the possibility of verification. It seems that the purpose is to analogize factors that are different from existing risk factors. The factors contained in the original paper are limited, and statistically, exploratory verification is limited. If possible, it is advisable to add an approximate number of these and potential factor candidates.
---

VERSION 1 – AUTHOR RESPONSE

Reviewer 1:

Comment:

none

Response:

Thank you for reviewing the protocol.

Reviewer comment 2:

Comment:

The protocol entitled, "Risk factors associated with gall bladder cancer in high incident areas in India: a systematic review protocol" presents an attempt to analyse risk factors for gallbladder cancer in a systematic review. I believe that it will be meaningful in research related to the epidemiology of carcinogenesis in specific regions and diseases.

It is planned according to PRISMA-P to search for risk factors. I confirmed that it was properly planned with the checklist. On the other hand, I don't think there is much mention of sample size, the number of target factors, and the possibility of verification.

It seems that the purpose is to analogize factors that are different from existing risk factors. The factors contained in the original paper are limited, and statistically, exploratory verification is limited. If possible, it is advisable to add an approximate number of these and potential factor candidates.

Response:

Thank you for your valuable comments. We are conducting a systematic review with the objective of identifying probable risk factor in the high-risk regions of India encompassing Ganga Brahmaputra River basins as there are no concrete evidence. To date, worldwide, there are approximately 15 types of factors that have been found to be associating gallbladder cancer from independent studies worldwide. For example, demographic (age, gender, socio-economic status, residence, education), anthropometric factors, nutritional factors (body weight, waist to hip ratio), reproductive factors (age at menarche, number of pregnancies, age at first child, age at marriage, number of childbirths, number of abortions, menstrual status, age at last childbirth, method of contraception, multiparity), exposure to environmental substances, typhoid/enteric infection, dietary and cooking practices and geographical factors. We have now mentioned in page 6 in the protocol, line#184-187, under the 'analysis of subgroups or subset' section.

As this is a systematic review, the sample size from individual studies will be extracted and for the meta-analysis (as mentioned in the protocol on page number 5, line#154, under 'data extraction' section).

Verification of factors unfortunately is beyond the scope of this review as we rely on the secondary data reported by the authors in their respective studies. However, we are extracting data on laboratory or other relevant methods, when available, for example, Vidal test and PCR data for detecting *Salmonella typhi*, as reported by the authors to assess heterogeneity in the sub-group analysis. We have now added this on page 5, line# 155, under the 'data extraction' section.

VERSION 2 – REVIEW

REVIEWER	Nakagawa, Kei Tohoku University, Surgery
REVIEW RETURNED	19-Dec-2021
GENERAL COMMENTS	The authors have responded appropriately to questions from editors and reviewers.